# Verification Study of Nanostructure Evolution with Heating Treatment between Thin and Thick Fullerene-Like Hydrogen Carbon Films

**Zhaolong Wang** [1]**, Kaixiong Gao** [1,2]**, Bin Zhang** [1,2,*]**, Zhenbin Gong** [1,2]**, Xiaoli Wei** [3] **and Junyan Zhang** [1,2]

[1] State Key Laboratory of Solid Lubrication, Lanzhou Institute of Chemical Physics, Chinese Academy of Sciences, Lanzhou 730000, China; wangzldyx@163.com (Z.W.); kxgao@licp.cas.cn (K.G.); gongzhb2011@163.com (Z.G.); zhangjunyan@licp.cas.cn (J.Z.)

[2] Center of Materials Science and Optoelectronics Engineering, University of Chinese Academy of Sciences, Beijing 100049, China

[3] School of Chemical Engineering, Lanzhou University of Arts and Science, Lanzhou 730000, China; weixli123@126.com

[*] Correspondence: bzhang@licp.cas.cn; Tel.: +86-931-4968191

**Abstract:** Fullerene-like hydrogen carbon films with a thin film grown on a NaCl substrate are usually employed to show the nanostructure of films (usually of hundred nanometers thick grown on Si substrates) under high resolution transmission electron microscopy (HRTEM) tests because it is easier floated off, where dependability and reasonability has never been seriously contested. Thus, in this paper, thin and thick hydrogen carbon films have been deposited on NaCl (thin films) and Si (thick films) substrates and annealed under room temperature to 500 °C, of which nanostructures have been investigated by HRTEM, Raman spectroscopy, and X-ray photoelectron spectroscopy, to verify the dependability and reasonability of the NaCl method. The results showed heating induced graphitization but with hydrogen content nearly unchanged. HRTEM results revealed that under annealing of 200, 250, and 300 °C, the curved graphene structures gradually increase in films. However, beyond 400 °C, onions structures are present. However, both Raman and XPS spectra show us that after annealed treatment, for original films, both thin and thick films have the near $sp^2$ bonding content and size, but with the annealing temperature increase, $sp^2$ bonding content increases more quickly for thick FL-C:H films due to the higher internal stress compared to thin films. In one word, the NaCl method used for nanostructure detection for films might be a good choice for an easier and quicker analysis, but it is still insufficient, because the heating effect induced by plasma cannot be ignored.

**Keywords:** carbon films; friction; annealed treatment

## 1. Introduction

Carbon, the most fascinating of elements, props up the whole living system and drives social progress. Not only that, the findings of new carbon materials (like carbon nanotubes, fullerenes, and graphene) provide new application chance to people, which promotes the progress of science and society. Nowadays, carbon-based materials are extensively explored to reduce friction, which consumes more than one third of energy that human's use every day. Some of the carbon materials are more attractive because of their superlubricity properties (friction coefficient <0.01). Superlubricity, was proposed by M. Hirano and K. Shinjo in 1991 [1], to define the statues of the friction coefficient <0.01 [2]. More than 100 papers have been devoted to study the mechanism of superlubricity, which is

due to incommensurate contacts between atomically smooth lattice planes at the microscale, such as molybdenum disulphide ($MoS_2$), highly oriented pyrolytic graphite (HOPG), and carbon nanotubes (MWCNT) [3–6]. The closest practical application of superlubrication is achieved by the introduction of carbon nanoscrolls [7–9], which can be considered as micro-incommensurate contacting. Formation of nanoscrolls from both out-situ [7] and in-situ [8] graphene structures is a good strategy to realize superlubricity under open air conditions. In contrast, fullerene-like hydrogen carbon (FL-C:H) films, which buckled by curved faulty-graphene planes (which lead to basal planes intersecting each other and the basal planes in this structure are interlocked by covalently linked tetrahedral sp3 bonds to form a three-dimensional network structure) are prominent candidates for industrial superlubricity due to in-stiu formation of nanoscrolls. Those faulty-graphenes peeling off from bulk of FL-C:H films during friction, which are believed to be the main routing to the formation of nanoscrolls [8]. That is to say, the more orderly of FL-C:H films, the more easily realized the superlubricity. Therefore, studying the growth mechanism and controlling growth of ordering FL-C:H films are important for scientists.

Until now, FL-C:H films can be grown using micro wave plasma CVD, reactive magnetron sputtering and high-power impulse-CVD etc. [10–13]. However, the growth mechanism is still under debate. Zhang and Wang [14] compared the studied nanostructure of hydrogen carbon films growth under different duty cycle of impulse power. Their study showed that it is possible to control the microstructure evolution of FL-C:H films via adjusting the bias on-off time and growth atmospheres. The result of Liu et al. [14] also confirmed that the long relaxation time contributed to the fullerene-like structure formation. However, one should notice that the fullerene-like carbon nitride (FL-CN) film caused growth on high temperature with low bias assistants. Inevitably, heating effects are described as booster graphitization [15]. In addition, the growth of FL-CN film was attributed to the N substitute of C which induced curvature at high temperature, which are determined via high resolution transmission electron microscopy (HRTEM) results [16].

Usually, such inner nanostructures are able to be distinguished by HRTEM, using samples grown on NaCl substrate [16] to help understanding the growth mechanisms of carbon-based films. However, the dependability and reasonability of the NaCl method has never been seriously contested.

To verify the NaCl method, in this paper, the thermal stability of the thin and thick FL-C:H films was studied by annealing in a nitrogen atmosphere under different temperatures. Elastic recoil detection (ERD), scanning electron microscopy (SEM), high resolution transmission electron microscopy (HRTEM), Raman spectra, and X-ray photoelectron spectroscopy (XPS) were employed to research the structure variation of the FL-C:H films.

## 2. Materials and Methods

### 2.1. Material Preparations

Before deposition, Si (100) substrates were cleaned for 30 min with ethanol in the ultrasonic cleaning device and were rapidly transferred into a vacuum chamber. When the internal pressure of the deposition chamber came down to $9.0 \times 10^{-4}$ Pa, FL-C:H films were beginning to be deposited with the gas (methane) pressure of 16 Pa, negative voltage of 870 V, pulsed frequency about 60 kHz, duty cycle of 0.6, and the deposition time was 60 min (thick films, and the deposition rate is 14.6 nm/min). In addition, the freshly cleaved NaCl wafers were employed for growth of thin films in the same condition and the deposition time was 60 s, and the deposition rate was 13.3 nm/min. The heat-treated FL-C:H film specimens were produced by heating at 200, 250, 300, 400, and 500 °C in an inert atmosphere (nitrogen) for 0.5 h with a heating rate of 4 °C/min and cooled down to room temperature in a chamber.

### 2.2. Characterization Methods

The elastic recoil detection (ERD) was used to explore the content of hydrogen in FL-C:H films. The 2.5 MeV He was used for a detection beam and the angle of incidence for the detection beam was 75° with respect to the normal to the sample surface. The protons recoiled from the films were

detected by a silicon surface barrier detector (was placed at 30° to the beam). A 13 μm Mylar foil was placed in front of the detector for protection from forward scattering He particles. Scanning electron microscopy (SEM, JSM-5601LV, JEOL, Tokyo, Japan), transmission electron microscopy (TEM, FEI Tecnai F30, FEI, Eindhoven, The Netherlands), nano-indentater DCM system, micro-Raman analyses (Jobin-Yvon HR 800 and HR Evolution, Horiba/JobinYvon, Longjumeau, France), X-ray photoelectron spectroscopy (XPS, operating with Al-Ka radiation and detecting chamber pressure of below $10^{-6}$ Pa) (Physical Electronics Inc., Chanhassen, MN, USA) were employed to reveal the microstructure variation of FL-C:H films. The TEM sample, about 20 nm thick film, was grown on freshly cleaved NaCl wafers (single crystals), and then dissolved with distilled water and placed on Cu grids for further testing. Prior the XPS detection, Au thin films about 0.2 nm thick were deposited on the tested carbon film surfaces, which minimized the charging effect, and helped to detect the changes of Cls peaks. The thicknesses of the thin and thick films about 80 and 870 nm are determined via TEM cross-section methods.

## 3. Result

### 3.1. The Results of ERD

All samples that were treated by tempering were transformed into an EDR chamber to detect the hydrogen content, and one of the results is shown in Figure 1a. All data are parsed, and the results of the hydrogen content are described in Figure 1b for thick films. One can see that the contents of hydrogen of the annealed samples is kept constant below 250 °C, though the content of hydrogen of the annealed samples decrease linearly with an increase of the treated temperature (250–500 °C), the changes can be generally ignored, which is only from 24.45% to 23.3%. The hydrogen content trends are similar with Li et al. [17]. The results of the thin film are nearly the same with a thick one that might be due to the accordance growth conditions. That means the release of hydrogen is not only determined by heating, at least in the present work, that heating treatment influence on the hydrogen content is insignificant [18], compared with the work of Manimunda et al. [19].

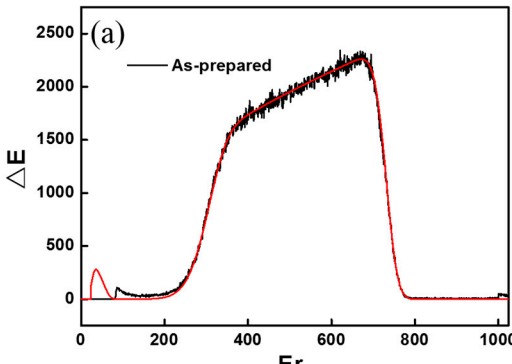 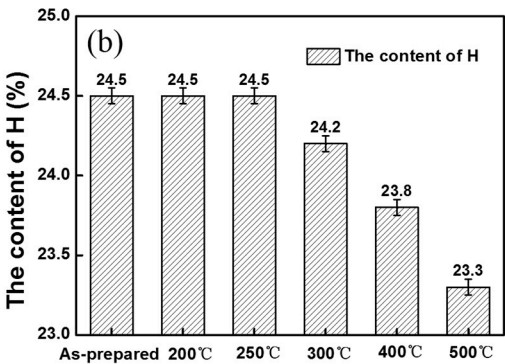

**Figure 1.** (**a**) Energy spectrum of the H component of FL-C:H films under different annealing conditions, (**b**) the content of H of FL-C:H films under different annealing conditions.

### 3.2. The Results of HRTEM

Figure 2a–f demonstrates the HRTEM plane-view images and the selected area electron diffraction (SAED) patterns of thin films under different annealing temperatures. Figure 2a is the picture of an untreated film and Figure 2b–f images are of samples annealed at 200, 250, 300, 400, and 500 °C, respectively. Curved graphene structures can be seen sparsely from as-prepared film. Though the as-prepared film annealed under low temperature of 200, 250, and 300 °C, one can observe that curved structures come distinctly as well as SAED patterns. The corresponding SAED patterns exhibit three fuzzy rings (marked as $a \approx 1.15$ Å, $b \approx 2.25$ Å, and $c \approx 3.5$ Å). The rings with 1.15, 2.25, and 3.5 lattice spacing correspond with $C_{60}$ and hexagonal basal planes of graphite (0002) and the spacing between

the carbon layers in bucky onion structures, respectively [13,14]. Beyond 400 °C, however, the onions structures are present, particularly for those annealed under 500 °C. Meanwhile, the SAED patterns of films under the temperature of 400 and 500 °C are much brighter.

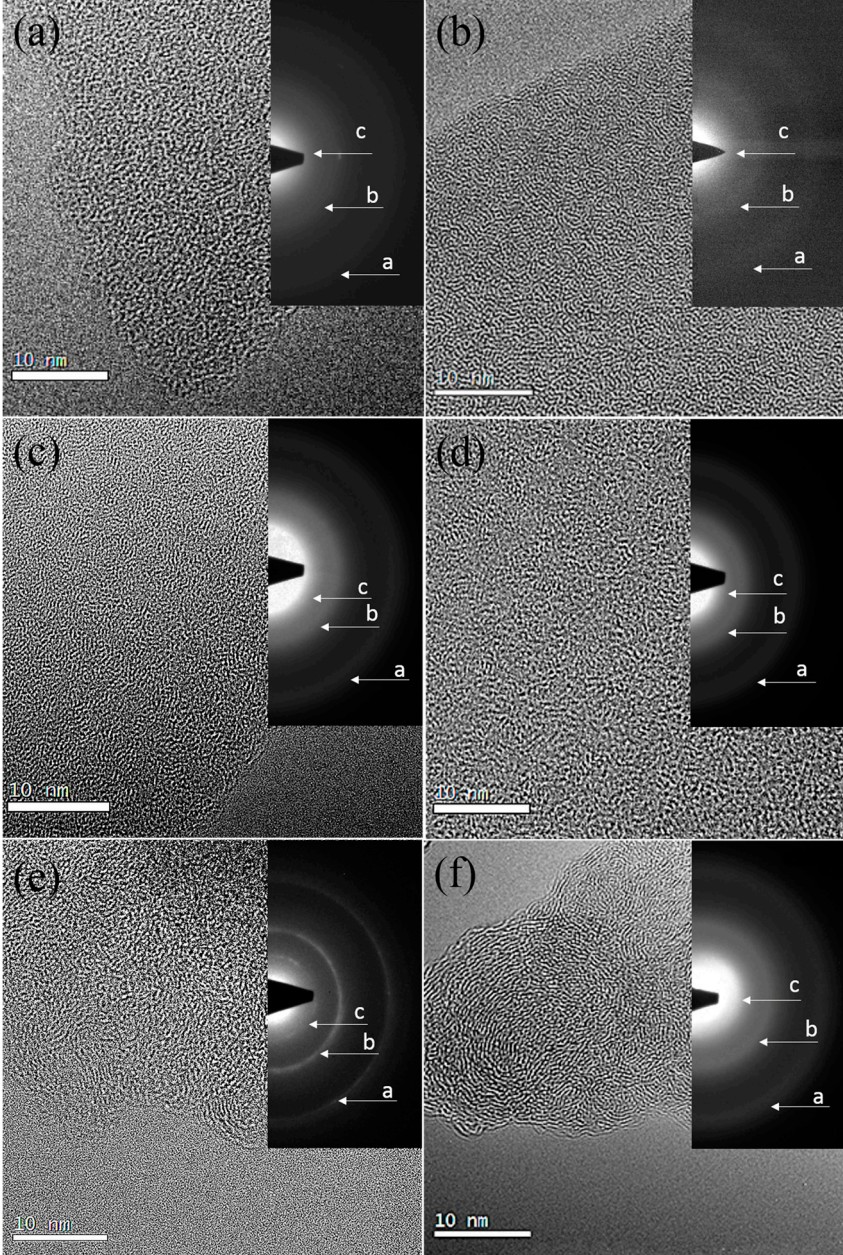

**Figure 2.** TEM images corresponding to the selected area electron diffraction pattern of FL-C:H films: (**a**) Is the image of the film, which is not subjected to the heat treatment, and the images of the annealed films at (**b**–**f**) of 200, 250, 300, 400, and 500 °C, respectively.

### 3.3. The Analysis of Raman Spectrums

Owing to outstanding sensitivity on the structure of $sp^2$ carbon bonds, Raman spectroscopy analysis is often used to study carbon-based films. Particularly, multi-wavelength Raman studies usually give more reliable results than single-wave length measurements [19]. Here, both thin and thick FL-C:H films before and after annealing are studied via Raman spectroscopy, with wavelengths of both 325 and 532 nm to verify the dependability and reasonability of the NaCl method.

One can find the Raman spectra of 325 nm as shown in Figure 3a,b for thin and thick films, respectively. The as prepared thin films give a weak peak at about 1600 cm$^{-1}$ (G peak), and no D peaks near 1380 cm$^{-1}$ can be observed. The G peak comes obviously with the increasing of treated temperature, the hump near 1380 cm$^{-1}$ arises for the sample annealed at 400 °C, and sharps of that sample were treated at 500 °C. The results are accordant with HRTEM. However, for thick films, both peaks near 1370 cm$^{-1}$ (D peak) and at 1600 cm$^{-1}$ (G peak) can be seen before annealing. With an increase of annealing temperature, D and G peak come distinctly, which is very similar with thin film results. However, there was just a little difference, in that no peaks near 1370 cm$^{-1}$ (D peak) presented below 300 °C for the thin films. The possible reason is that after a long time, growth from heat accumulation cannot be ignored, that is, the graphitization happening due to the releasing of stress can form more sp$^2$ structures with an increase in growth time. More details can be found in Reference [14].

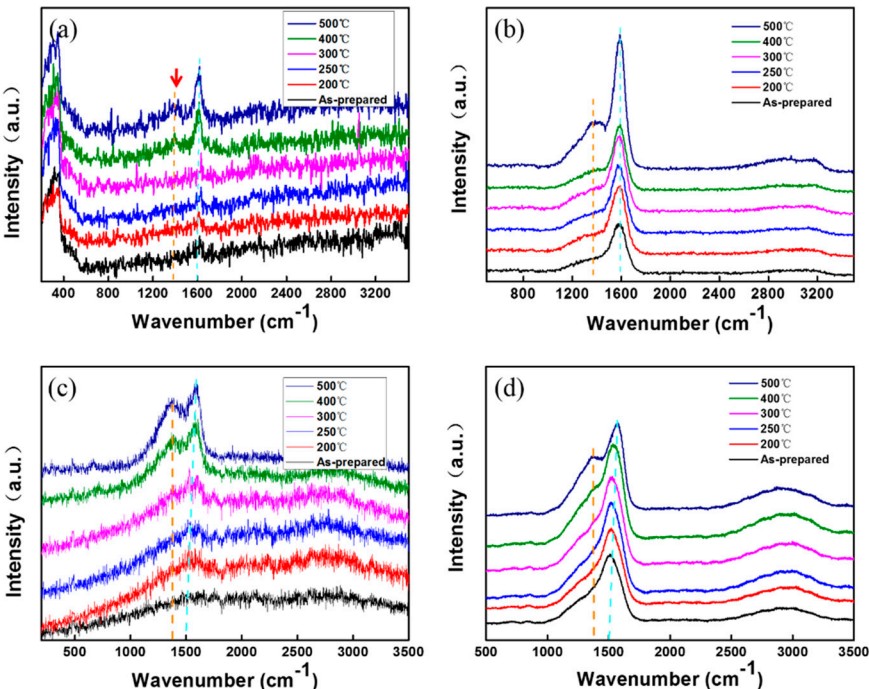

**Figure 3.** Raman spectra of thin (**a**) and thick (**b**) FL-C:H films in different annealing temperatures for λL = 325 nm, Raman spectra of thin (**c**) and thick (**d**) FL-C:H films in different annealing temperatures for λL = 532 nm.

To test the speculation, Raman spectra of wavelength under 532 nm are analyzed. The results are shown in Figure 3c,d. For the as-prepared thin film, different from that of wavelength under 325 nm, no peaks can be found because of the noise background below 300 °C. However, with the increase of the annealed temperature, both peaks near at 1360 (D peak) and 1570 cm$^{-1}$ (G peak) come obviously and sharpen, and the red shift of the G Peak is observed. For the as-prepared thick film, a sharp G peak can be seen with a D shoulder. With the increasing of the treated temperature, not only the red shift of the G Peak is noticed, but also the distinct of the D peak can be seen.

In one word, both Raman spectra of wavelengths under 532 and 325 nm provide us with information that the thin film coated on NaCl are different from the thick film deposited on Si(100). With those treated below 300 °C, no D peak can be found with a distinct G peak for the thin film. However, both the D and G peak can be seen from the as-deposited thick film to annealing under different temperature thick ones. The assured reason is that the releasing of stress can form more sp$^2$ structures [14].

Moreover, the average size of sp$^2$ clusters can be obtained by the following Equation:

$$L_a = \sqrt{\frac{I_D / I_G}{\acute{C}(\lambda)}} \tag{1}$$

where there is a scaling coefficient (0.55 nm$^{-2}$ for the Raman excitation wavelength of 532 nm) and La is the planar dimension of the clusters given in nm units [18]. Figure 4 shows the average size changes of sp$^2$ clusters for the thin and thick films. It is observed that the common trend of sp$^2$ clusters for the thin and thick films grows with increasing temperature. However, the performance of the thin films slows down.

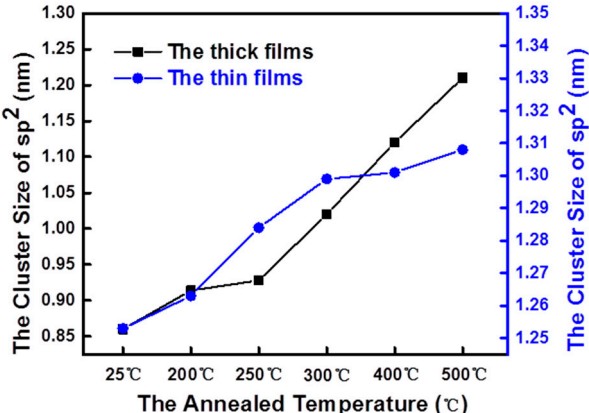

**Figure 4.** The average size changes of sp$^2$ clusters for the thin and thick films.

In one word, both Raman spectra of wavelengths under 532 and 325 nm provide us with information that the thin film was coated on NaCl, where we always employ analysis with nanostructures that are not exactly the same with the thick one.

### 3.4. The Results of XPS

X-ray photoelectron spectroscopy (XPS) is the most effective technology of films surface component analysis [20,21]. Because each element has a unique electronic structure, XPS can carry out a study of the basic chemical structure of solid surface materials. The C1s core-level XPS spectra of thin and thick FL-C:H films are shown in Figure 5. It can be seen clearly from the picture that the C1s binding energies of the original and heat-treated FL-C:H films (thin and thick) are in the range of 284.4–284.7 eV. These values can be appraised by fitting the C1s peak with three components (Figure 6a), graphite represented by the sp$^2$ bond (peak at 284.3 eV), diamond represented by the sp$^3$ bond (peak at 285.1 eV), and detect represented by the C–O bond (peak at 286.3 eV), and by calculating the area fractions of the peaks [22]. As shown in Figure 5, compared with thin films, the XPS of thick films own the narrow full width at half maxima (FWHM). Moreover, the XPS peak of thick films shifts to low binding energy at a temperature of 500 °C in comparison with thin films. It means that thick films have more ordering structure than the thin ones.

Furthermore, the bonding structures were studied by means of XPS-peak-differentiation-imitating analysis, the values of sp$^2$/sp$^3$ were calculated on thin films and thick films according to the fitted results of the XPS spectra (Figure 6b). It can be seen that the variation tendency of sp$^2$/sp$^3$ increase both thin and thick films from room temperature to high annealed temperature (500 °C). However, the thick FL-C:H films own higher values of sp$^2$/sp$^3$ after annealing treatment, which implies the higher level of graphitization compared with thin films [23]. The possible reason is that the thick films have higher internal stress than thin ones, and the annealing process to the change of stress field for the formation of graphene-like or fullerene-like structures, more details can be found in our previous work [14].

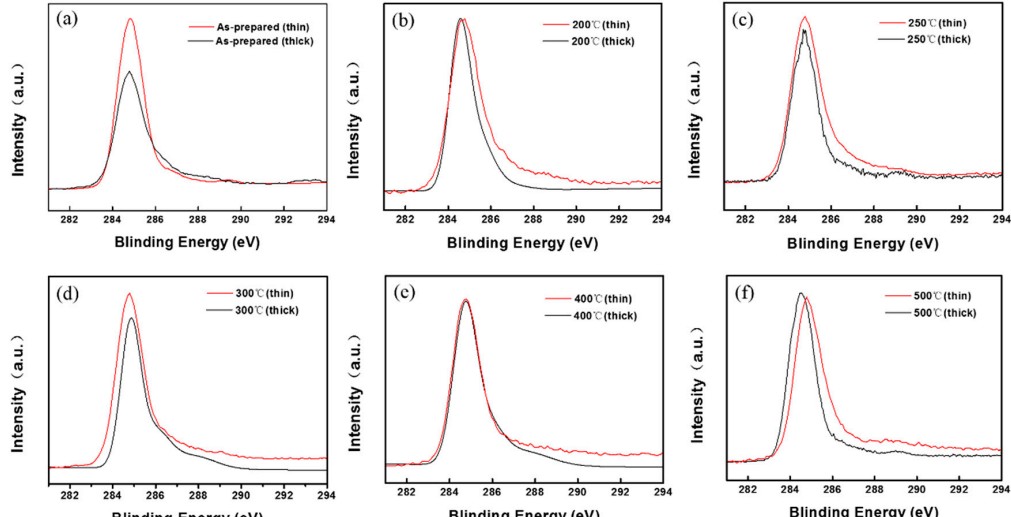

**Figure 5.** The XPS spectra of thick and thin FL-C:H films, (**a**–**f**) is original films and heat treatment films with annealed temperatures of 200, 250, 300, 400, and 500 °C, respectively.

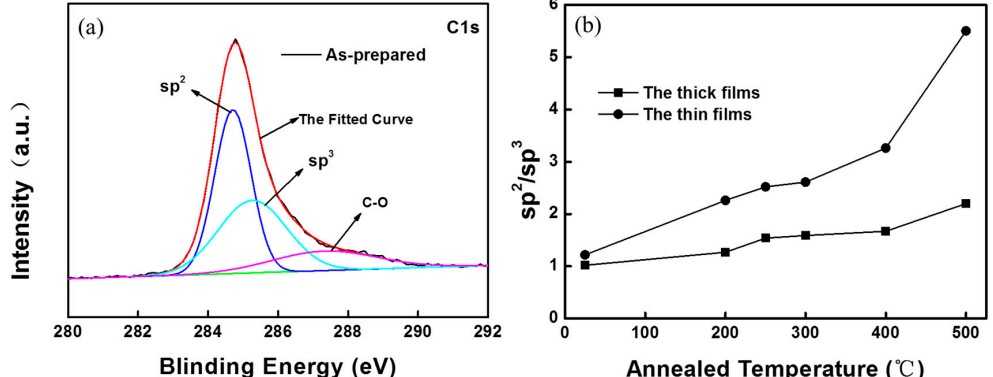

**Figure 6.** The fitted XPS C1s of original FL-C:H films (**a**), the values of $sp^2/sp^3$ for thin and thick FL-C:H films at different annealed temperatures (**b**).

In addition, the value of C/O for the films (thin and thick) was also provided according to the XPS spectra (Figure 7). For the thin films, the C/O was lowest at a temperature of 300 °C, and the highest at 500 °C. However, for the thick films, the C/O was always raised from the original state to the annealed one of 500 °C.

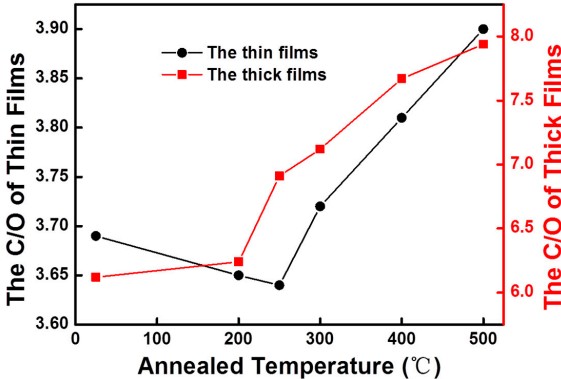

**Figure 7.** The value of the C/O for the thin and thick films at different annealed temperature.

## 4. Discussion

Thin films grown on NaCl wafers are frequently used as a guide sample to study the nanostructure via the HRTEM method, but no one disputes its reliability. Considering the distinct thickness, and always grown on different substrates (NaCl for a thin one because it is easier to transfer, and Si for the thick one), the verification study for the NaCl methods for nanostructures' dependability and reasonability has never been carried out in the present work.

From the HRTEM results, one can conclude that with an annealed temperature increase, the graphitization becomes more drastic, but is still different from Li et al. [16] and Manimunda et al. [17], with not much change in hydrogen content. It implies that the fullerene-like structure has more chemical durability than amorphous ones. One can speculate that the hydrogen atoms are transferred to the edge of the graphene-based clusters, thus the nanostructure is more ordering than the original one, which is proved by HRTEM.

However, HRTEM results of thin films reveal the true variation of the thick one. To verify this, both thin and thick FL-C:H films were studied via Raman and XPS methods to verify the dependability of the HRTEM results. As shown in Figure 3, no matter which wavenumber of Raman was chosen, the evolution trends of the thin and thick films were the same, which were also confirmed by XPS. Both Raman and XPS results show the graphitization trends and grow up of $sp^2$ clusters for thin and thick FL-C:H films, in accordance with HRTEM tests. However, for Raman spectra, the original peak shape is difference from the thin and thick FL-C:H films, that no D peaks are found from unheated samples for wavelengths under 325 and 532 nm for thin ones but are clearly seen for thick ones. However, similar Raman spectra shapes turn out beyond 300 °C, that means the bonding structure has greater convergence at high annealing temperatures. For XPS, the variation of thin and thick films is the same with each other, but obviously the thick films were easier to transform into a $sp^2$ cluster (Figures 4 and 6) [24]. As shown in Figures 4 and 6, for the original films, both thin and thick films have the near $sp^2$ bonding content and size, but with the annealing temperature increase, $sp^2$ bonding content increased more quickly for thick FL-C:H films due to the higher internal stress compared thin films [14].

## 5. Conclusions

The heat treatment of the thin and thick FL-C:H films deposited by the PECVD systems were investigated in this article. The chief opinions are as below:

- The content of hydrogen of the annealed samples remained constant, thus, the influence of hydrogen could be negligible in this study.
- Under low temperature annealing of 200, 250, and 300 °C, the curved graphene structures gradually increased in films. However, beyond 400 °C, onions structures were present, particularly if annealed under 500 °C.
- Both Raman and XPS spectra show us that after the annealed treatment, for original films, both thin and thick films have the near $sp^2$ bonding content and size, but with the annealing temperature increase, $sp^2$ bonding content increases more quickly for thick FL-C:H films due to the higher internal stress compared to thin films

In one word, the NaCl method used for nanostructure detection for films might be a good choice for an easier and quicker way for analysis, but it is still insufficient, because the heating effect induced by plasma cannot be ignored.

**Author Contributions:** Conceptualization, B.Z., K.G. and Z.W.; Methodology, Z.W.; Software, Z.W. and B.Z.; Validation, G.Z., K.G. and X.L.; Formal Analysis, B.Z.; Investigation, Z.W.; Resources, X.W. and J.Z.; Data Curation, Z.W.; Writing-Original Draft Preparation, Z.W.; Writing-Review & Editing, Z.W. and B.Z.; Supervision, J.Z.; Project Administration, B.Z.; Funding Acquisition, J.Z.

**Funding:** This work is supported by Youth Innovation Promotion Association CAS (No. 2017459), Science and Technology Service Network Initiative CAS and the National Natural Science Foundation of China (Nos. U1737213, 5181101666) and Natural Science Foundation of Gansu Province (No. 18JR3RA233).

**Acknowledgments:** We thank the support of Science and Technology Service Network Initiative CAS for this article. And we also thank Andre Anders very much for helping ERD analysis.

**Conflicts of Interest:** The authors declare no conflict of interest.

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
