# Peer review of "Verification Study of Nanostructure Evolution with Heating Treatment between Thin and Thick Fullerene-Like Hydrogen Carbon Films"

_coatings, doi:10.3390/coatings9020082_

Round 1
Reviewer 1 Report
In the present manuscript entitled “Verification study nanostructure evolution with heating treatment between thin and thick fullerene-like hydrogen carbon films” the authors present their results of deposited fulleren-like hydrogen carbon films (FL-C:H) on two substrates, where one is coated with thin films, the NaCl, and one with thick films, Si-wafers. With heat treatment the composition and structures of the films change, which the author documented by several analytical methods.
The organization of the manuscript itself is in a good order to see the developing characteristics and especially the used analytics, but it can’t be read fluently. The English used by the authors needs improvement, e.g., use of tenses, singular vs. plural, but also several sentences are ambiguous and a clear statement not understandable (see some examples attached). Therefore I highly recommend to get some help of a science Anglophone to assist in the reediting of the paper. In the current status it is not publishable.
Additionally, there are some points, which should be addressed in a revised manuscript before publication. Therefore, I would recommend major revision of the manuscript based on the following comments:
-The authors use thin and thick films. The authors never mentioned the thicknesses in dimension (nm, µm). The thin ones (e.g., 20 nm for TEM, but the rest???) are applied on NaCl (cubic system) as substrate and the thick ones on Si (diamond structure): Why do they compare thin films with thick films on different substrates, and not same thicknesses? Are the growth mechanisms the same for both substrates?
-Is there any substrate effect detectable in the carbon-layer?
-Is there any stress effect in the layers, which is released by the heating and forms more sp2 bondings in the layers by forming a denser network with simultaneous release of Hydrogen as demonstrated for higher temperatures by ERD?
-It is well known that with different resp. increasing thickness the composition of the deposited carbon films can change also in H-content especially for PECVD methods (compare: Catena et. al.: ACS Appl. Mater. Interfaces 2016, 8, 10636-10646. ). This also has to be taken into account by comparing thin and thick films with analytical methods like XPS, which are surface sensitive (How deep is your penetration depth?). So in my opinion a simple comparison between these thin and thick layers is not possible. Therefore, the manuscripts misses a detailed discussion or the chosen comparison has to be explained in more detail.
-What are the characteristics for FL-C:H? The authors explained this in their cited references, but it should be mentioned briefly here as well. Instead, the Raman or XPS can also be from an amorphous type of C:H. How can the authors prove the FL-C:H especially on the thick layers used for Raman and XPS. Here again what is the thickness of the thick layers on Si. Are there any problems with reflectivity of the Si samples.
-In the discussion the graphitization of the layers is mentioned. What is the cluster size of Sp2 regions? Therefore, the use of an optical spectroscopy resp. an absorbance spectroscopy of the carbon layers to check the Tauc gap (Tg) of the film and furthermore the estimation of cluster size for sp2 configurations etc. would be helpful (compare: Robertson, Adv. Phys. 35 (1986) 317; Clausing, NATO ASI Ser. B Phys. 266 (1991) 331; Ferrari, Phys. Rev. B 61 (2000) 14095, Catena, Carbon 2017, 117, 351)
-The morphology resp. AFM images of the films would be as well interesting for the “nanostructure” verification
-Why is the N-doping of FL-C:H layers mentioned in the introduction, is there any need for later? I do not see a point for N-doped layers …as it is just an heating in inert air (here N2)
-More information on the specific analytics resp. used parameters is needed
-Raman for sure is a widely used method for these layers, but no references or examples are given herein
-In Figure 5: the red line is missing in the legend
-What is the conclusion why annealing causes more sp2 content in the FL-C:H layers. The authors present a lot analytics but no explanation for the phenomenon. A clear discussion resp. conclusion is here missing.
Some sentences, only a selection, there are far more, where rework is needed to clarify the statement:
-Line 20 etc.: However, though the XPS results display that sp2 content of hydrogen is accordance well with each other for thin and thick original FL-C:H films and separate with increased temperature,
-Line 33: …superlubricy properties…???
-Line 41 etc. However, in contrast, thanks to the unique structures of fullerene-like hydrogen carbon (FL-C:H), which can provide faulty-graphene via peeling off of which from bulk films during friction to formation nanoscrolls:
-Line 46: …compared studied the nanostructure of hydrogen carbon films growth under different impulse power
-Line 59: … gonging…???
-Line 68: Then conclude that there are differences between thin and thick FL-C:H films under heat treatment, but still have a certain referential significance-making it easier for scientists to study. – What does this mean?
-Line 127: … and to verify that thin film on NaCl wafer can be actual response the truth of thick one.
-Line 135: … extra-hoop can be find from as-prepared films … - what is an extra-hoop in an Raman spectrum
-A.s.o.
Author Response
Dear Sir/Madam,
Thank you very much for your careful and kind.
We do modified the paper with your suggestion and mark it in Yelllow, Please considering our modification and give more advises. Thank you again for your help.
Bin

Reviewer 2 Report
The authors fabricated a carbon thin and thick film on Si and NaCl substrate by plasma CVD and evaluated by XPS, Raman, TEM and the like. Improvement is desirable before publication as follows Although it can be understood that using NaCl substrate for the convenience of TEM sample preparation, why was Si used as a substrate for control experiments? A more detailed explanation is desirable The solvent in ultrasonic cleaning of Si substrate not described Base pressure of plasma CVD and the degree of vacuum during film formation are not described The deposition rate is not described and is the same film deposition rate between Si substrate and NaCl substrate? There is no information on EDR in the experimental section What is the peak around 350 cm -1 observed at 325 nm excitation in the Raman spectrum? There are C - O peaks when peak fitting with C 1s in XPS, but to what extent is the C / O ratio varies with temperature?
Author Response
Dear Sir/Madam,
Thank you very much for your good suggestion.
Accordingly, We modified our paper to fit the Coatings requirement.
Thank you very much.
Bin

Round 2
Reviewer 1 Report
Dear authors,
the current manuscript entitled “Verification study nanostructure evolution with heating treatment between thin and thick fullerene-like hydrogen carbon films” has improved very much in the revised version. There are some minor typos in formatting etc, which can be corrected via the editorial process or in the proof. Overall it reads sound and clear.
The additional infornmation and graphics support much the findings, so I recommend acception.
Reviewer 2 Report
I am satisfied with revised version. I recommend to publish after the correction that garbled characters of formula (1) on line 172 of Page 6.